# Mono-Alkyl-Substituted Phosphinoboranes (HRP–BH$_2$–NMe$_3$) as Precursors for Poly(alkylphosphinoborane)s: Improved Synthesis and Comparative Study

**Felix Lehnfeld [1], Tim Oswald [2], Rüdiger Beckhaus [2] and Manfred Scheer [1,\*]**

[1] Institut für Anorganische Chemie, Universität Regensburg, 93040 Regensburg, Germany; felix.lehnfeld@chemie.uni-regensburg.de

[2] Institut für Chemie, Carl von Ossietzky Universität Oldenburg, Carl-von-Ossietzky Straße 9–11, 26129 Oldenburg, Germany; tim.oswald@uni-oldenburg.de (T.O.); ruediger.beckhaus@uni-oldenburg.de (R.B.)

\* Correspondence: manfred.scheer@ur.de

**Abstract:** A new synthetic pathway to various mono-alkyl-substituted phosphinoboranes HRP–BH$_2$–NMe$_3$ has been developed. The new synthetic route starting from alkyl halides and NaPH$_2$ followed by metalation and salt metathesis is performed in a one-pot procedure and leads to higher yields and purity of the resulting phosphinoboranes, as compared to previously reported routes. Additionally, the scope of accessible compounds could be expanded from short-chained linear alkyl substituents to longer-chained linear alkyl substituents as well as secondary or functionalized alkyl substituents. The reported examples include primary alkyl-substituted phosphinoboranes RHP-BH$_2$-NMe$_3$ (R = *n*-butyl, *n*-pentyl, *n*-hexyl; **1a–c**), the secondary alkyl-substituted derivatives *i*PrPH-BH$_2$-NMe$_3$ (**2**), and the functionalized alkyl-substituted 4-bromo-butyl-phosphinoborane (BrC$_4$H$_8$)PH-BH$_2$-NMe$_3$ (**3**). Compounds **1a**, **1c,** and **2** were additionally used for preliminary polymerization reactions via a thermal and a transition metal-catalyzed pathway, revealing the formation of high-molecular-weight polymers under certain conditions. Detailed investigations on the influence of temperature, concentration, substituents and reaction time on the respective polymerization reactions were performed.

**Keywords:** 13/15 compounds; boron; phosphorus; polymerization

## 1. Introduction

Polymers are an integral part of our everyday lives, not only as plastics for daily use; they also play an essential role in industry[1,2]. In addition to their plethora of useful material properties, the control of these properties is particularly crucial for their successful application. The structure within the material can be altered via modification after polymerization, or by modifying the starting material before polymerization. Thus, the properties of a polymer can be controlled [3,4].

In addition to organic polymers, inorganic main group polymers have also attracted increased attention due to a large number of specialized applications such as for ceramic or luminescent materials or in optoelectronics [5–23]. Common synthetic routes towards such polymers are based on polycondensation or ring-opening polymerization processes. Of special interest are transition metal-based dehydrocoupling reactions, as they can lead to high-molecular-weight polymers under rather mild conditions [24–35]. Polymers based on group 13 and 15 elements are an important class of compounds accessible via dehydrocoupling reactions [34–44]. Due to the nonpolar nature of P-H bonds, usually, electron-withdrawing aryl substituents on the phosphorus atom are necessary for such

reactions. Examples of alkyl-substituted polymers obtained through dehydrocoupling are limited [39,41,44].

Our group was able to synthesize alkyl-substituted phosphorus-boron polymers through different procedures; the thermal elimination of the stabilizing Lewis base in phosphinoboranes of the type RHP-BH$_2$-NMe$_3$ (R = *t*Bu, **I**; R = Me, **II**, Figure 1) leads to the formation of high-molecular-weight polymers [45,46]. Via this pathway, the formation of alkyl-substituted arsenic-boron oligomers has also been reported recently [47]. Moreover, the titanium-catalyzed polymerization of **I** under very mild conditions and with shorter reaction times was achieved (Figure 1) [48,49]. As yet, the scope of accessible polymers has been limited by the range of suitable starting materials, as only a few mono-alkyl-substituted phosphinoboranes have been reported so far; these are essential for accessing a broader variety of properties. Therefore, the search for a more generally applicable and improved synthetic route to alkyl-substituted phosphinoboranes is still ongoing.

**Figure 1.** Reported examples of the polymerization of alkyl-substituted phosphinoboranes.

Herein, we report on a new synthetic procedure for the synthesis of alkyl-substituted phosphinoboranes which is applicable to a variety of primary alkyl substituents and to secondary and even functionalized alkyl residues. Furthermore, the obtained monomers were used as starting materials for the polymerization reactions initially studied. In particular, the influence of various conditions was investigated in detail.

## 2. Results and Discussion

### 2.1. General Synthetic Procedure

So far, alkyl-substituted phosphinoboranes have been synthesized via two different synthetic routes: either via the metalation of a phosphine and subsequent salt metathesis reaction (Figure 2, path **A**) [46], or via the formation of a phosphonium-borane salt by the reaction of PH$_2$-BH$_2$-NMe$_3$ with alkyl halides and subsequent deprotonation (Figure 2,

path **B**) [45]. However, both methods have their disadvantages. Path **A** works very well for commercially available phosphines such as *t*BuPH$_2$, but otherwise a prior synthesis of the phosphine is necessary. The established laboratory scale synthesis of monoalkylated phosphines may be considered elaborate, time-intensive and expensive. Additionally, the overall reaction is a rather waste-intensive process, and has a low atom efficiency.

Method **B** has so far only been reported for short-chained, primary alkyl substituents (R = Me, Et, *n*Pr), and is therefore rather limited in scope. Furthermore, the yield of the resulting phosphinoboranes is mediocre at best when moving beyond the methyl-substituted derivative (Figure 2).

**Figure 2.** Previously reported syntheses of monoalkyl-substituted phosphinoboranes. Path **A**: salt metathesis method, path **B**: quarternization and deprotonation method.

In view of other possible reaction pathways, the synthesis of several monoalkyl phosphines from PH$_3$ and alkyl halides was reported [50]. However, the challenge of handling the highly toxic phosphine gas in a laboratory makes this method difficult for the synthesis of substituted phosphines as starting material for phosphinoboranes on a day-to-day basis. By replacing the phosphine gas with NaPH$_2$ and performing the alkylphosphine synthesis and the subsequent metalation of the phosphine by NaNH$_2$ in a one-pot procedure, the release of toxic gases can be reduced to a minimum. This step can be followed up by a simple salt metathesis of the resulting phosphanide NaRHP with IBH$_2$NMe$_3$, leading directly to the desired phosphinoborane (Figure 3). Thus, with all starting materials readily available in gram scale and good yields, the products are accessible on a large scale.

**Figure 3.** General synthetic route for monoalkyl-substituted phosphinoboranes starting from NaPH$_2$, alkylhalides and IBH$_2$NMe$_3$.

Via this route, different phosphinoboranes can be obtained in medium-to-good yields as colorless oils. In some cases, synthesis and workup at low temperature are necessary to prevent unwanted thermal polymerization.

All obtained compounds revealed a broad singlet or pseudo-quartet in the $^{31}$P{$^1$H} NMR spectra, which showed further splitting into a triplet in the $^{31}$P NMR spectra. The chemical shifts for phosphinoboranes with primary alkyl substituents are about δ = −130 ppm, for an *i*Pr substituent at δ = −92 ppm. Independent of the substituents, all compounds

revealed similar $^1J_{P,H}$ coupling constants of about 200 Hz. In the $^{11}B\{^1H\}$ NMR spectra, a doublet for all compounds was observed with very similar chemical shifts around δ = −10 ppm and a $^1J_{P,B}$ coupling constant of about 50 Hz. In the $^{11}B$ NMR spectra, all products revealed further splitting into a triplet of doublets with $^1J_{B,H}$ couplings of ca. 100 Hz. All observed chemical shifts are in good agreement with the values for already-reported phosphinoboranes [45,46,51]. The most prominent side product observed was the dialkyl-substituted phosphinoborane, which can be suppressed by slow addition of the alkyl halide to $NaPH_2$ at a low temperature.

Some general trends were perceived; the synthesis of primary alkyl phosphinoboranes proceeded with good yields, with the *n*-butyl- and *n*-hexyl-substituted compounds being the highest, at 55%. Furthermore, while the formation of the most prominent side product, dialkyl-substituted phosphinoborane, was not observed for primary alkyl-substituted phosphanylboranes, a yield of about 15% for the *i*Pr-substituted derivative could be substantiated. A similar decrease in yield and purity was observed for the formation of a bromoalkyl-substituted phosphanylborane.

## 2.2. Phosphinoboranes with Primary Alkyl Substituents

Expanding on the scope of the already reported $MeHP-BH_2-NMe_3$ derivative, further phosphinoboranes with primary alkyl substituents were investigated. To cover a variety of different features within this type of substituents, three different phosphinoboranes were prepared: the rather short-chained $nPrHP-BH_2-NMe_3$ (**1a**), the longer-chained $nHexHP-BH_2-NMe_3$ (*n*Hex = *n*-hexyl; **1c**) and the linear isomer to the well-investigated *t*Bu derivative, $nBuHP-BxH_2-NMe_3$ (**1b**). All compounds were accessible via adjusted synthetic procedures of method **B,** starting from the unsubstituted $H_2P-BH_2-NMe_3$, but yields were rather moderate (for details, refer to Materials and Methods, and Supplementary Materials). Using the new method reported herein, the yields could be heavily increased, especially considering the phosphorus-based atom efficiency. Additionally, the reaction can be scaled up significantly compared to previous reports, offering improved access to these monomers as precursors for poly(phosphinoborane)s. The NMR chemical shifts as well as the yields of the compounds **1a–c** obtained via this method and via adjusted literature methods are summarized in Table 1.

**Table 1.** Chemical shifts and yields of compounds **1a–c** obtained using adjusted literature procedures and as reported in this work.

| Compound | Substituent | δ ($^{31}P$) [a] | δ ($^{11}B$) [a] | Yield [b] |
|:---:|:---:|:---:|:---:|:---:|
| **1a** | *n*-propyl | −130.4 | −3.2 | 28%/29% |
| **1b** | *n*-butyl | −127.6 | −4.0 | 13%/54% |
| **1c** | *n*-hexyl | −128.7 | −3.3 | 32%/66% |

[a] in ppm in toluene solution; [b] adjusted literature procedure [46]/this work; both referring to $NaPH_2$.

### Preliminary Investigations of **1a–c** as Polymer Precursors

Polymerization experiments with compounds **1a–c** were performed under varying conditions, both via thermal elimination of the stabilizing $NMe_3$ Lewis base as well as via catalytic conditions using the recently reported bispentafulvene $[(\eta^5:\eta^1-C_5H_4C_{10}H_{14})_2Ti]$ ([Ti]) catalyst [49,52]. Overall, the influence of temperature, reaction time, concentration and solvent as well as the presence and concentration of the catalyst were investigated to be discussed in the following.

For all three compounds, thermal polymerization at roomtemperature reveals comparably long reaction times. Under these conditions, several days of stirring are necessary to achieve a conversion of >50% of the starting material. Reaction times of more than two weeks lead to increasing amounts of decomposition as observed by new signals in the $^{11}$B and $^{31}$P NMR spectra, without any noteworthy increase in conversion rates. Already slightly elevated temperatures (323 K) lead to a significant decrease in the necessary reaction times, resulting in similar conversion rates in solution after only three to five days.

The concentration of the phosphinoborane has a strong influence on its polymerization behavior. In all cases, already noticeably shorter reaction times lead to full conversion for more concentrated systems. The best results were achieved when almost no solvent was used. Due to the increasing viscosity of the formed polymers, traces of toluene are always necessary to still achieve full conversion; therefore, a completely neat approach is not feasible. When the phosphanylborane was heated in the presence of traces of toluene to 323 K, full conversion to a clean polymer was achieved within only two days of stirring. A clear trend emerged during the studies: The chain length of the substituent directly influences the reaction times; thus, with increasing chain length, reaction times also increased. Changing the solvent to more polar solvents such as THF or using mixtures of toluene and THF did not impact the reaction in a meaningful way.

When the polymerization reaction was performed in the presence of the recently reported bispentafulvene [Ti] catalyst [49,52], significant differences could be observed: The presence of the [Ti] catalyst leads to drastically reduced reaction times in the range of hours, but in all cases, the reaction stops at around 80% conversion. In this case, neither an increase of temperature nor longer reaction times lead to an improvement. Differences in catalyst loading, identified as an important factor in the [Ti]-catalyzed polymerization of *t*BuHPBH$_2$NMe$_3$, do not lead to any observable differences for the investigated reactions with compounds **1a–c**. Along the lines of the thermal polymerization, the concentration of the phosphinoborane seems to be crucial for good conversions and fast reactions. The most significant reduction in the reaction time accompanied by higher conversion was observed when only traces of toluene were used as solvent. In contrast to the thermal polymerization of **1-c** or the catalytic polymerization of *t*BuPHBH$_2$NMe$_3$, however, complete conversion was not observed for **1a–c** under the applied conditions in the presence of the [Ti] catalyst.

From the primary alkyl-substituted monomers, the polymer obtained from **1c** was selected to be characterized by ESI+ mass spectrometry in addition to multinuclear NMR spectroscopy. Whereas for the catalytic polymerization, only aggregates with a molar mass of up to 1700 Da (n = 12) could be detected, it was possible to observe peaks for polymers with up to 2700 Da (n = 20) for polymerization under thermal conditions (323 K, traces of toluene).

In both cases, polymers with three different end groups could be identified (Figure 4). Further investigation of the nature of these polymers considering polydispersity, tacticity and more detailed data on the end groups and chain lengths of the resulting polymers will be the focus of future work to give valuable insight into the nature of the resulting polymers.

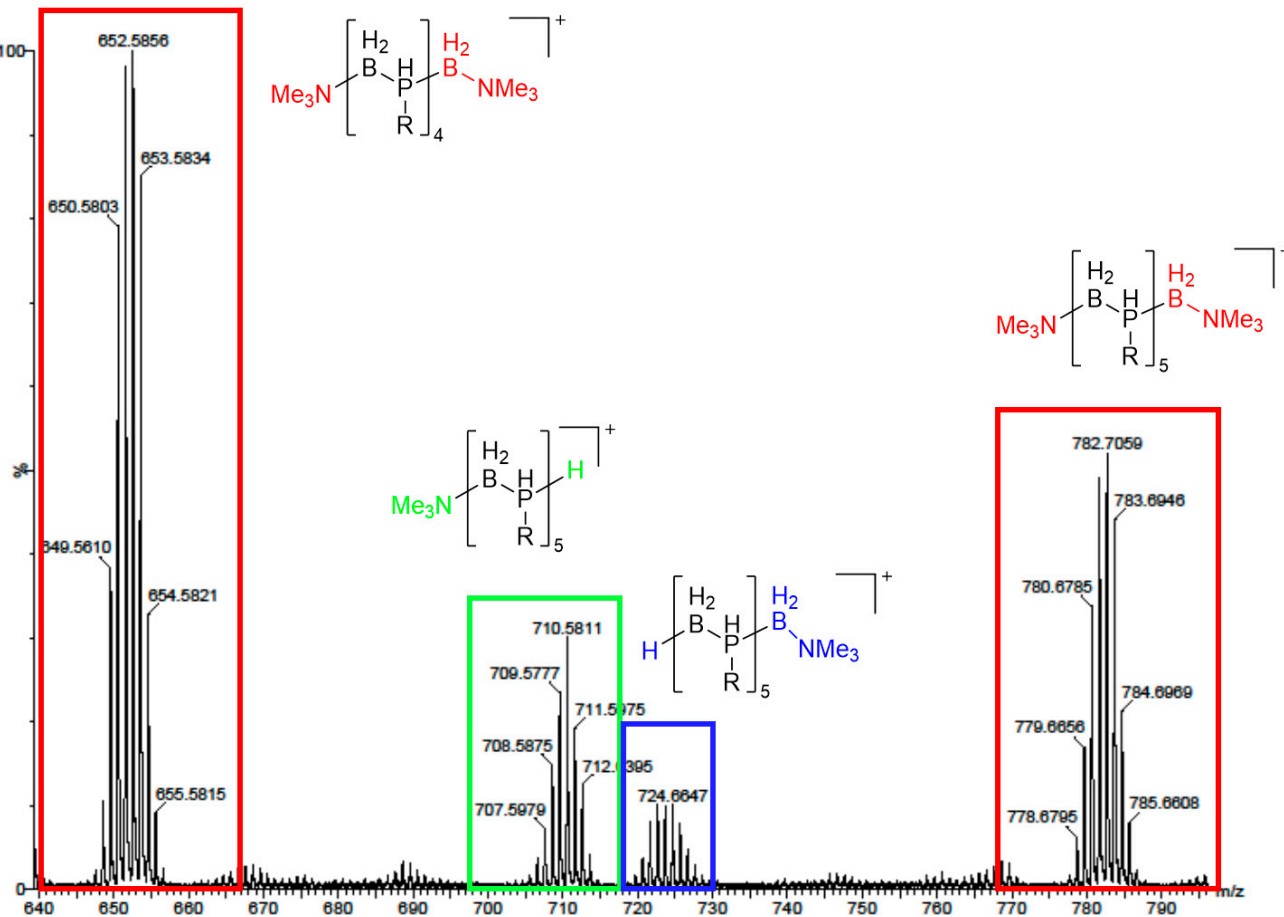

**Figure 4.** End groups observed in the ESI+ mass spectra of poly-**1c.**

### 2.3. Phosphinoboranes with Secondary Alkyl Substituents

Phosphinoboranes bearing a secondary alkyl substituent should position themselves as a promising starting material in terms of solubility, reactivity, and steric hinderance between phosphinoboranes with primary alkyl substituents such as *n*Pr and tertiary alkyl substituents such as *t*Bu. However, the quest for a suitable phosphinoborane with such a secondary alkyl substituent is still ongoing, as the synthesis via method **B** for such substituents is not applicable, as reported. Adapting the reported method **B** using one equivalent of the Lewis acid AlCl₃ to activate the iso-propyl halide led to the formation of both the phosphonium borane salt and the corresponding phosphinoborane after a secondary deprotonation step.

Applying the synthetic procedure presented in this work allowed for a significant improvement; the additional Lewis acid was not needed, while improved purity and good yields were provided (Figure 5). The reaction showed almost no side reactions or decomposition, even less than those observed for **1a–c**. The obtained phosphinoborane was characterized using multinuclear NMR spectroscopy, as discussed in the first part of the Results and Discussion section.

**Figure 5.** Synthesis of an *i*Pr-substituted phosphanylborane **2.**

In contrast to other phosphinoboranes, the *i*Pr-substituted phosphinoborane **2** revealed a considerably lower tendency for polymerization. Thermal polymerization at room temperature only led to minimal conversion, while [Ti]-based catalytic polymerization revealed an unexpected behavior; similar to what was reported for the parent phosphinoborane [49], the formation of polymeric species was not observed. Instead, a significant broadening of the signal corresponding to **2** in the $^{31}$P NMR spectra was perceived, which increased with higher catalyst load without any new signals emerging (Figure 6). Simultaneously, an unusual color change to a deep purple was observed, indicating the potential formation of a paramagnetic species. In the case of a stochiometric reaction with the [Ti] complex, the signal for compound **2** disappeared completely from the NMR spectra, but unfortunately, no product could be isolated, regardless of numerous attempts.

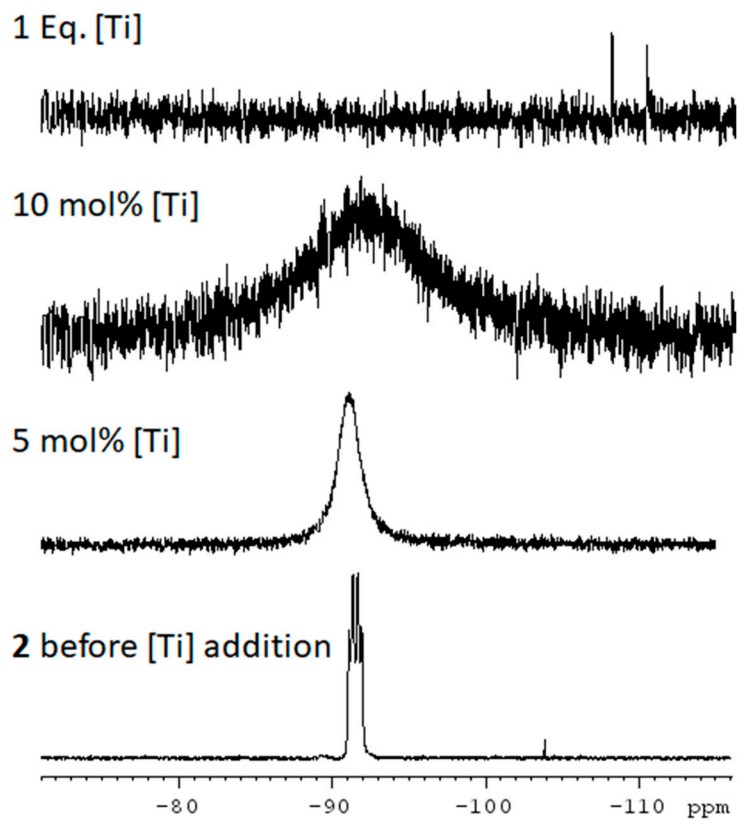

**Figure 6.** Signal broadening in the $^{31}$P{$^1$H} NMR spectra of the reaction solution of **2** in toluene in the presence of different amounts of [Ti] at 293 K.

Its unusual behavior in both the thermal and catalytic polymerization experiments makes **2** a very interesting compound for future investigations, as both isolating a product from the reaction with the [Ti] catalyst and applying different conditions for the thermal polymerization, such as using more concentrated solutions as well as elevated temperatures, led to more insights.

### 2.4. Phosphinoboranes with Functionalized Alkyl Substituents

In the industrial application of polymers, crosslinking and other linked networks play an important role. Therefore, the investigation of phosphinoboranes with additional functionalized groups is of great interest.

Using well-developed phosphinoboranes with primary alkyl substituents allows us to access this kind of chemistry. Crosslinking should be possible by preforming the linkage in the monomer. Through the reaction of NaPH$_2$ with an alkyldihalide with

suitable chain length, a diphosphine and a subsequent diphosphinoborane are made accessible (Figure 7).

The first step of the reaction yields the corresponding diphosphine. The reaction proceeds with full conversion of the crude reaction mixture according to $^{31}$P NMR spectroscopy. In addition to the diphosphine, revealing a triplet at δ = −137.3 ppm in the $^{31}$P NMR spectrum ($^1J_{P,H}$ = 189 Hz), bromobutylphosphine BrC$_4$H$_8$PH$_2$ is formed as a minor side product. Upon metalation and reaction with IBH$_2$NMe$_3$, a very unselective reaction is observed, but in addition to unreacted diphosphine, two promising products can be identified: the bridged diphosphinoborane **3a** and the phosphinebutylphosphinoborane **3b**, which only reacted with one equivalent of NaNH$_2$. A product ratio of 1:1 was found.

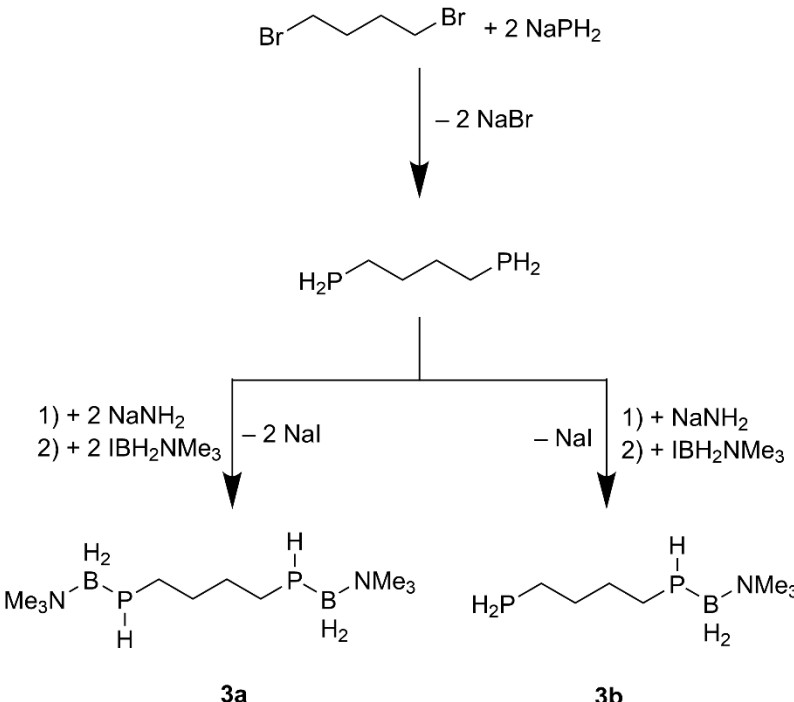

**Figure 7.** Synthesis of a butyl-bridged diphosphinoborane **3a** and the most prominent side product **3b**.

The two products were investigated via NMR spectroscopy. Compound **3a** reveals a pseudo-quartet at δ = −68.8 ppm in the $^{31}$P{$^1$H} NMR spectrum, whereas for **3b**, a similar signal is observed at δ = −128.1 ppm. In both cases, coupling to the boron atom was observed with $^1J_{P,B}$ coupling constants of 51 Hz. Both show further splitting as a doublet of multiplets in the $^{31}$P NMR spectrum with $^1J_{P,H}$ coupling constants of approx. 44 Hz. For **3b**, the signal for the unreacted PH$_2$ group was identified as a triplet at δ = −138.7 ppm with a $^1J_{P,H}$ coupling constant of 189 Hz. In the $^{11}$B{$^1$H} NMR spectrum, **3a** revealed a broad doublet at δ = −3.9 ppm and **3b** a strongly broadened signal at δ = 4.5 ppm. In the $^{11}$B NMR spectrum, both revealed further splitting with $^1J_{B,H}$ coupling constants of 120 Hz (**3a**) and 112 Hz (**3b**), respectively. In addition, multiple decomposition products can be detected in the $^{11}$B NMR spectrum. Therefore, further tuning of the reaction conditions is necessary to isolate clean diphosphinoborane from this reaction procedure, as purification of this reaction mixture was not possible within the scope of this work.

### 3. Materials and Methods

#### 3.1. General Remarks

All reactions were performed in an argon or nitrogen inert-gas atmosphere using standard glove-box and Schlenk techniques. All solvents were taken from a solvent purification system of the type MB-SPS-800 from the company MBRAUN( Garching, Germany), and degassed via standard procedures. All nuclear magnetic resonance (NMR) spectra were recorded on a Bruker Avance 400 spectrometer (Brucker Instruments, Ettlingen, Germany) ($^1$H: 400.13 MHz, $^{13}$C{$^1$H}: 100.623 MHz, $^{11}$B: 128.387 MHz, $^{31}$P: 161.976 Hz) with δ [ppm] referenced to external standards ($^1$H and $^{13}$C{$^1$H}: SiMe$_4$, $^{11}$B: BF$_3$-Et$_2$O, $^{31}$P: H$_3$PO$_4$). All mass spectra were recorded on a Micromass LCT ESI-TOF (Waters Corporation, Wexford, Irland).

#### 3.2. Synthesis of nPrPHBH₂NMe₃ (**1a**), nBuPHBH₂NMe₃ (**1b**), and nHexPHBH₂NMe₃ (**1c**) by Adjusted Literature Procedures

Following the literature procedure for MePHBH$_2$NMe$_3$ [46], compounds **1a–1c** were prepared by using *n*PrI, *n*BuI and *n*HexI as a replacement for MeI.

Yields (referred to NaPH$_2$)

**1a**: 72 mg (0.49 mmol, 28%).
**1b**: 57 mg (0.46 mmol, 13%).
**1c**: 125 mg (0.67 mmol, 38%).

#### 3.3. One-Pot Synthesis of nPrPHBH₂NMe₃ (**1a**), nBuPHBH₂NMe₃ (**1b**), nHexPHBH₂NMe₃ (**1c**)

For the preparation of **1b**, an H-shaped Schlenk flask for condensation was filled with NaPH$_2$ (0.99 g, 17.6 mmol) on one side and NaNH$_2$ (700 mg, 17.9 mmol) on the other side. NaPH$_2$ was suspended in 5 mL tetrahydrofurane (THF) and the suspension cooled to 213 K. *n*BuI (2 mL, 3.24 g, 17.6 mmol) was added to the suspension and the mixture stirred for 16 h, while the solution was allowed to reach room temperature. The formation of a white precipitate was observed. The resulting THF solution of *n*BuPH$_2$ was then condensed under reduced pressure onto the solid NaNH$_2$ in the other half of the flask, which was cooled with liquid nitrogen. The mixture was allowed to reach room temperature under stirring. After stirring overnight, a yellow solution was obtained. A solution of IBH$_2$NMe$_3$ (3.8 g, 17.6 mmol) in THF was added to the yellow solution, and the mixture stirred overnight. A color change from yellow to colorless and the formation of a white precipitate were observed. After removing the solvent in vacuo, compound **1b** was extracted with *n*-hexane and filtered over diatomaceous earth. After removing the solvent in vacuo, compound **1b** was obtained as a colorless oil.

Compounds **1a** and **1c** were obtained via the same procedure, using *n*PrI and *n*HexI instead of *n*BuI, and with slightly different sample sizes.

Yield:

**1a**: m = 0.414 g (2.82 mmol, 29%)
**1b**: m = 1.537 g (9.5 mmol, 54%)
**1c**: m = 1.256 g (6.6 mmol, 66%)

**1a**: $^{31}$P{$^1$H} NMR (toluene, 293 K): δ [ppm] = −130.4 (q, $^1J_{P,B}$ = 46 Hz); $^{31}$P NMR (toluene, 293 K): δ [ppm] = −130.4(dq, $^1J_{P,H}$ = 206 Hz, $^1J_{P,B}$ = 48 Hz); $^{11}$B {$^1$H} NMR (C$_6$D$_6$, 293 K): δ [ppm] = −3.2 (d, $^1J_{P,B}$ = 48 Hz); $^{11}$B NMR (C$_6$D$_6$, 293 K): δ [ppm] = −3.2 (td, $^1J_{B,H}$ = 107 Hz, $^1J_{P,B}$ = 48 Hz).

**1b**: $^1$H NMR (C$_6$D$_6$, 293 K): δ [ppm] = 0.96 (t, $^3J_{H,H}$ = 7 Hz, 3H, nBu-CH$_3$), 1.53 (m, 2H, nBu-CH$_2$), 1.63–1.88 (m, 4H, nBu-CH$_2$), 1.94 (s, 9H, NMe$_3$), 2.43 (m, $^1J_{P,H}$ = 196 Hz, 1H, PH), 2.25–3.10 (q, 2H, $^1J_{B,H}$ = 105 Hz, BH$_2$); $^{31}$P{$^1$H} NMR (C$_6$D$_6$, 293 K): δ [ppm] = −127.7 (q, $^1J_{P,B}$ = 46 Hz); $^{31}$P NMR (C$_6$D$_6$, 293 K): δ [ppm] = −127.7(dq, $^1J_{P,H}$ = 196 Hz, $^1J_{P,B}$ = 46 Hz); $^{11}$B {$^1$H} NMR (C$_6$D$_6$, 293 K): δ [ppm] = −3.9 (d, $^1J_{P,B}$ = 46 Hz); $^{11}$B NMR (C$_6$D$_6$, 293 K): δ [ppm] = −3.9 (td, $^1J_{B,H}$ = 105 Hz, $^1J_{P,B}$ = 46 Hz).

**1c**: $^1$H NMR (C$_6$D$_6$, 293 K): δ [ppm] = 0.88 (t, $^3J_{H,H}$ = 7 Hz, 3H, *n*Hex-CH$_3$), 1.33 (m, 2H, *n*Hex-CH$_2$), 1.54 (pent, 2H, *n*Hex-CH$_2$), 1.64–1.90 (m, 4H, *n*Hex-CH$_2$), 1.95 (s, 9H, NMe$_3$), 2.47 (m, $^1J_{P,H}$ = 198 Hz, 1H, PH), 2.25–3.10 (q, 2H, $^1J_{B,H}$ = 105 Hz, BH$_2$); $^{31}$P{$^1$H} NMR (C$_6$D$_6$, 293 K): δ [ppm] = −128.2 (q, $^1J_{P,B}$ = 48 Hz); $^{31}$P NMR (C$_6$D$_6$, 293 K): δ [ppm] = −128.2 (dq, $^1J_{P,H}$ = 198 Hz, $^1J_{P,B}$ = 48 Hz); $^{11}$B {$^1$H} NMR (C$_6$D$_6$, 293 K): δ [ppm] = −4.1 (d, $^1J_{P,B}$ = 48 Hz); $^{11}$B NMR (C$_6$D$_6$, 293 K): δ [ppm] = −4.1 (td, $^1J_{B,H}$ = 105 Hz, $^1J_{P,B}$ = 48 Hz).

### 3.4. Synthesis of [iPrPH$_2$BH$_2$NMe$_3$]AlCl$_3$I

To a solution of 253 mg (1.9 mmol) AlCl$_3$ in 10 mL Et$_2$O, a solution of 199 mg (1.9 mmol) PH$_2$BH$_2$NMe$_3$ in 4 mL toluene was added. After stirring for 15 min at r.t., 0.19 mL *i*PrI (1.9 mmol) was added and the mixture was stirred for 40 h. All volatiles were removed in vacuo and a colorless oil was obtained. The product was washed with toluene (2 × 5 mL) at −30 °C and extracted with 2 × 5 mL THF and filtrated over diatomaceous earth. After removing the solvent in vacuo, a colorless oil containing [*i*PrPHBH$_2$NMe$_3$]AlCl$_3$I was obtained. Two side products could not be further identified. Further purification was not possible; therefore, no exact yield could be determined.

$^{31}$P{$^1$H} NMR (toluene, THF, 293 K): δ [ppm] = −64.4 (q, $^1J_{P,B}$ = 65 Hz); 31P NMR (toluene, THF, 293 K): δ [ppm] = −64.4 (t, $^1J_{P,H}$ = 416 Hz); $^{11}$B{$^1$H} NMR (toluene, THF, 293 K): δ [ppm] = −11.6 (m); $^{11}$B NMR (toluene, THF, 293 K): δ [ppm] = −11.6 (m).

### 3.5. Synthesis of iPrPHBH$_2$NMe$_3$ (**2**)

#### 3.5.1. From [iPrPHBH$_2$NMe$_3$]AlCl$_3$I

To a solution of 51 mg (0.475 mmol) lithiumdiisopropylamide (LDA) in 5 mL THF, a solution of the product mixture of [*i*PrPHBH$_2$NMe$_3$]AlCl$_3$I in 3 mL CH$_3$CN was added at −243 K. After stirring for 30 min at 243 K, all volatiles were removed in vacuo at 243 K. The product was extracted with 3 × 4 mL *n*-hexane and filtrated over diatomaceous earth. After removing the solvent in vacuo, **2** was obtained as a colorless oil. This oil could not be further purified; thus, no yield could be determined.

#### 3.5.2. From a One Pot Synthesis

For the preparation of **2**, an H-shaped Schlenk flask for condensation was filled with NaPH$_2$ (560 mg, 10 mmol) on one side and NaNH$_2$ (390 mg, 10 mmol) on the other side. NaPH$_2$ was suspended in 5 mL THF and the suspension cooled to 213 K. *i*PrI (2 mL, 760 mg, 10 mmol) was added to the suspension and the mixture stirred for 16 h, while the solution was allowed to reach room temperature. The formation of a white precipitate was observed. The resulting THF solution of *i*PrPH$_2$ was then condensed onto the solid NaNH$_2$ in the other half of the flask. The mixture was allowed to reach room temperature under stirring. After stirring overnight, a yellow solution was obtained. A solution of IBH$_2$NMe$_3$ (2.0 g, 10 mmol) in THF was added to the yellow solution and the mixture was stirred overnight. A color change from yellow to colorless and the formation of a white precipitate was observed. After removing the solvent in vacuo, compound **2** was extracted with n-hexane and filtered over diatomaceous earth. After removing the solvent in vacuo, compound **2** was obtained as a colorless oil.

Yield: 529 mg (3.6 mmol, 36%). $^{31}$P{$^1$H} NMR (toluene, 293 K): δ [ppm] = −92.5 (q, $^1J_{P,B}$ = 49 Hz); $^{31}$P NMR (toluene, 293 K): δ [ppm] = −92.5 (dq, $^1J_{P,H}$ = 203 Hz, $^1J_{P,B}$ = 49 Hz); $^{11}$B{$^1$H}-NMR (toluene, 293 K) δ = −4.3 (d, $^1J_{P,B}$ = 49 Hz); $^{11}$B-NMR (toluene, 293 K) δ = −4.3 (td, $^1J_{P,H}$ = 203 Hz, $^1J_{P,B}$ = 49 Hz).

### 3.6. Synthesis of Me$_3$NBH$_2$PH$_2$C$_4$H$_8$PH$_2$BH$_2$NMe$_3$ (**3a**) and H$_2$PC$_4$H$_8$PH$_2$BH$_2$NMe$_3$ (**3b**)

For the preparation of **3a** and **3b**, an H-shaped Schlenk flask for condensation was filled with NaPH$_2$ (560 mg, 10 mmol) on one side and NaNH$_2$ (390 mg, 10 mmol) on the other side. NaPH$_2$ was suspended in 20 mL THF and the suspension cooled to 213 K. 1,4-dibrombutane (0.59 mL, 1.1 g, 5 mmol) was added to the suspension and the mixture

stirred for 16 h, while the solution was allowed to reach room temperature. The formation of a white precipitate was observed. The resulting THF solution was then condensed onto the solid $NaNH_2$ in the other half of the flask. The mixture was allowed to reach room temperature under stirring. After stirring for 4 h while slowly warming up to room temperature, a yellow solution was obtained. A solution of $IBH_2NMe_3$ (2.0 g, 10 mmol) in THF was added to the yellow solution and the mixture was stirred overnight. A color change from yellow to colorless and the formation of a white precipitate was observed. After removing the solvent in vacuo, compounds **3a** and **3b** were extracted with *n*-hexane and filtered over diatomaceous earth. After removing the solvent in vacuo, the mixture of **3a** and **3b** was obtained as a colorless oil.

**3a**: $^{31}P\{^1H\}$ NMR (toluene, 293 K): δ [ppm] = −68.8 (q, $^1J_{P,B}$ = 51 Hz); $^{31}P$ NMR (toluene, 293 K): δ [ppm] = −68.8 (dq, $^1J_{P,H}$ = 192 Hz, $^1J_{P,B}$ = 51 Hz); $^{11}B\{^1H\}$-NMR (toluene, 293 K) δ = −3.9 (d, $^1J_{P,B}$ = 51 Hz); $^{11}B$-NMR (toluene, 293 K) δ = −3.9 (t, $^1J_{H,B}$ = 120 Hz).

**3b**: $^{31}P\{^1H\}$ NMR (toluene, 293 K): δ [ppm] = −128.1 (q, $^1J_{P,B}$ = 51 Hz, **PH$_2$BH$_2$**), −138.7 (s, **PH$_2$CH$_2$**); $^{31}P$ NMR (toluene, 293 K): δ [ppm] = −128.1 (dq, $^1J_{P,H}$ = 203 Hz, $^1J_{P,B}$ = 51 Hz), −138.7 (t, $^1J_{P,H}$ = 189 Hz, **PH$_2$CH$_2$**); $^{11}B\{^1H\}$-NMR (toluene, 293 K) δ = 4.5 (br); $^{11}B$-NMR (toluene, 293 K) δ = −4.5 (t, $^1J_{H,B}$ = 112 Hz).

### 3.7. Polymerization Experiments

#### 3.7.1. Of **1a**

A solution of **1a** in toluene was stirred under different conditions, either in the presence or absence of $[(\eta^5:\eta^1\text{-}C_5H_4C_{10}H_{14})_2Ti]$ ([Ti]). Details are summarized in Table 2.

**Table 2.** Conditions used in polymerization experiments of **1a**, catalyst: $[(\eta^5:\eta^1\text{-}C_5H_4C_{10}H_{14})_2Ti]$ ([Ti]).

| Catalyst Loading | Reaction Time | Temperature | Concentration 1a [mol/L] | Conversion |
|---|---|---|---|---|
| - | 90 min | r.t. | 0.089 | 19% |
| **-** | 24 h | r.t. | 0.089 | 23% |
| 5 mol% | 90 min | r.t. | 0.089 | 41% |
| 5 mol% | 24 h | r.t. | 0.089 | 51% |
| 10 mol% | 24 h | r.t. | 0.089 | 64% |
| 10 mol% | 210 min | r.t. | 0.03 | 54% |
| 10 mol% | 42 h | r.t.. | 0.03 | 76% |

#### 3.7.2. Of **1b** under Catalytic Conditions

A solution of **1b** (9.5 mmol mmol, 1.54 g) in 19 mL toluene was added to $[(\eta^5:\eta^1\text{-}C_5H_4C_{10}H_{14})_2Ti]$ (0.475 mmol, 211 mg, 5 mol%, [Ti]) and stirred for 21 d at room temperature. In the presence of [Ti], a color change to first green and then brown was observed. The solvent was removed in vacuo, and the remaining highly viscous dark brown oil was washed three times with cold *n*-pentane.

Conversion: 90%.

$^{31}P\{^1H\}$ NMR (toluene, 293 K): δ [ppm] = −63.3 (br, poly-**1b**); $^{31}P$ NMR (toluene, 293 K): δ [ppm] = −63.3 (br, poly-**1b**); $^{11}B\{^1H\}$ NMR (toluene, 293 K) δ = −38.0 (br, poly-**1b**); $^{11}B$ NMR (toluene, 293 K) δ = −38.0 (br, poly-**1b**).

#### 3.7.3. Of **1c**

A solution of **1c** in toluene was stirred under different conditions, either in the presence or absence of $[(\eta^5:\eta^1\text{-}C_5H_4C_{10}H_{14})_2Ti]$ ([Ti]). Details are summarized in Table 3. In the presence of [Ti], a color change to first green and then brown was observed.

**Table 3.** Conditions used in polymerization experiments of **1c**, catalyst: [(η$^5$:η$^1$-C$_5$H$_4$C$_{10}$H$_{14}$)$_2$Ti] ([Ti]).

| Catalyst Loading | Reaction Time | Temperature | Concentration 1c [mol/L] | Conversion |
|---|---|---|---|---|
| - | 16 h | r.t. | 0.1 | 18% |
| - | 16 h | 323 K | Neat [a] | 84% |
| - | 40 h | 323 K | Neat [a] | 97% |
| 10 mol% | 3 h | r.t. | 0.1 | 17% |
| 10 mol% | 3 h | r.t. | 0.2 | 27% |
| 4 mol% | 30 min | r.t. | 0.4 | 18% |
| 4 mol% | 90 min | r.t. | 0.4 | 22% |
| 4 mol% | 7 d | r.t. | 0.4 | 41% |
| 4 mol% | 21 d | r.t. | 0.4 | 65% |

[a] Traces of toluene/*n*-hexane are necessary to provide suitable viscosity during the reaction.

### 3.7.4. Of **2**

A solution of **2** in toluene (c = 0.2 mol/L) was stirred for 90 min either in the absence or in the presence of [(η$^5$:η$^1$-C$_5$H$_4$C$_{10}$H$_{14}$)$_2$Ti] (5 mol% or 10 mol%, [Ti]), and monitored via $^{31}$P NMR spectroscopy. In the presence of [(η$^5$:η$^1$-C$_5$H$_4$C$_{10}$H$_{14}$)$_2$Ti] ([Ti]), an unusual color change to purple was observed, having turned to brown after several days.

To further investigate this system, a solution of *i*PrPHBH$_2$NMe$_3$ (29 mg, 0.2 mmol) in 0.27 mL toluene was added to a solution of [(η$^5$:η$^1$-C$_5$H$_4$C$_{10}$H$_{14}$)$_2$Ti] (90 mg, 0.2 mmol, [Ti]) in 3 mL toluene. A color change to first blue-green, then green–purple and ultimately red-brown was observed. The solution was stirred for 16 h at r.t. and reduced to 0.75 mL. By precipitating in 40 mL CH$_3$CN, a gray-green solid was obtained. However, all attempts to isolate intermediates or characterize the products of this reaction have failed up to now.

### 4. Conclusions

Substituted phosphinoboranes are important precursors for functionalized inorganic polymers. Therefore, having a flexible synthetic route to producing phosphinoboranes with a variety of different substituents is of great value. With the new approach presented in this work, various primary alkyl-substituted phosphinoboranes are accessible in high purity and yields. The resulting compounds RPHBH$_2$NMe$_3$ (**1a–c**, R = *n*Pr, *n*Bu, *n*Hex) were used as starting materials for thermal and [Ti] catalyzed polymerization reactions. All three compounds revealed promising properties in these polymerizations, leading to the formation of polymers with up to 30 repetition units. These systems allowed us to investigate in detail the influence of various factors such as concentration, temperature, reaction time, and catalyst loading. Thus, it was possible to identify the best conditions for these polymerizations at elevated temperatures in very concentrated solutions, mostly independent of the catalyst concentration.

Furthermore, two new phosphinoborane derivatives were investigated: a phosphinoborane with an *i*Pr substituent as an example of a secondary phosphino-alkyl residue, and a diphosphinoborane system bridged by a butyl chain. They were synthesized via the reaction of the respective alkylhalides, isopropyliodide or dibromobutane with NaPH$_2$, followed by a subsequent metalation with NaNH$_2$ and additional salt metathesis with IBH$_2$NMe$_3$. In addition to being inaccessible via established synthetic routes up to date, both the *i*Pr-substituted phosphinoborane **2** and the butyl-bridged diphosphinoborane **3a** revealed surprising and interesting properties distinguishing them from already reported compounds. The detailed investigation of the features of these compounds as well as their potential as precursors for polymers will be the focus of future work.

**Supplementary Materials:** The following supporting information can be downloaded at: https://www.mdpi.com/article/10.3390/inorganics11100377/s1, Document S1. Figure S1: $^{31}$P NMR (top) and $^{31}$P$\{^1$H$\}$ NMR (bottom) spectra of **1a** in n-hexane, Figure S2: $^{11}$B NMR (top) and $^{11}$B$\{^1$H$\}$ NMR (bottom) spectra of **1a** in n-hexane, Figure S3: $^1$H NMR spectrum of **1b** in CD$_3$CN, Figure S4: $^{31}$P NMR (top) and $^{31}$P$\{^1$H$\}$ NMR (bottom) spectra of **1b** in CD$_3$CN, Figure S5: $^{11}$B NMR (top) and $^{11}$B$\{^1$H$\}$ NMR (bottom) spectra of **1b** in CD$_3$CN, Figure S6: $^1$H NMR spectrum of 1c in C$_6$D$_6$, Figure S7: $^{31}$P NMR (top) and $^{31}$P$\{^1$H$\}$ NMR (bottom) spectra of **1c** in C$_6$D$_6$, Figure S8: $^{11}$B NMR (top) and $^{11}$B$\{^1$H$\}$ NMR (bottom) spectra of **1c** in C$_6$D$_6$, Figure S9: $^{31}$P NMR (top) and $^{31}$P$\{^1$H$\}$ NMR (bottom) spectra of **2** in n-hexane, Figure S10: $^{11}$B NMR (top) and $^{11}$B$\{^1$H$\}$ NMR (bottom) spectra of **2** in n-hexane, Figure S11: $^{31}$P NMR (top) and $^{31}$P$\{^1$H$\}$ NMR (bottom) spectra of **3a** and **3b** in THF, Figure S12: $^{11}$B NMR (top) and $^{11}$B$\{^1$H$\}$ NMR (bottom) spectra of **3a** and **3b** in THF, Figure S13: $^{31}$P$\{^1$H$\}$ NMR spectra of **1a** (c = 0.089 mol/L) after stirring at r.t. for 90 min (bottom) and 24 h (top), Figure S14: $^{11}$B$\{^1$H$\}$ NMR spectra of **1a** (c = 0.089 mol/L) after stirring at r.t. for 90 min (bottom) and 24 h (top), Figure S15: $^{31}$P$\{^1$H$\}$ NMR spectra of **1a** (c = 0.089 mol/L) after stirring at r.t. for 90 min (bottom) and 24 h (top) in the presence of 5 mol% of [($\eta^5$ :$\eta^1$ -C$_5$H$_4$C$_{10}$H$_{14}$)$_2$Ti], Figure S16: $^{11}$B$\{^1$H$\}$ NMR spectra of **1a** (c = 0.089 mol/L) after stirring at r.t. for 90 min (bottom) and 24 h (top) in the presence of 5 mol% of [($\eta^5$ :$\eta^1$ -C$_5$H$_4$C$_{10}$H$_{14}$)$_2$Ti], Figure S16: $^{31}$P$\{^1$H$\}$ NMR and $^{11}$B$\{^1$H$\}$ NMR (top) spectra of **1a** (c = 0.089 mol/L) after stirring at r.t. for 24 h in the presence of 10 mol% of [($\eta^5$ :$\eta^1$ -C$_5$H$_4$C$_{10}$H$_{14}$)$_2$Ti], Figure S17: $^{31}$P NMR (top) and $^{31}$P$\{^1$H$\}$ NMR (bottom) spectra of **1a** (c = 0.03 mol/L) after stirring at r.t. for 210 min (lower half) and 42 h (upper half) in the presence of 10 mol% of [($\eta^5$ :$\eta^1$ -C$_5$H$_4$C$_{10}$H$_{14}$)$_2$Ti], Figure S18: $^{11}$B NMR (top) and $^{11}$B$\{$ $^1$H$\}$ NMR (bottom) spectra of **1a** (c = 0.03 mol/L) after stirring at r.t. for 210 min (lower half) and 42 h (upper half) in the presence of 10 mol% of [($\eta^5$ :$\eta^1$ -C$_5$H$_4$C$_{10}$H$_{14}$)$_2$Ti], Figure S19: $^{11}$B NMR (top, upper half), $^{11}$B$\{^1$H$\}$ NMR (bottom, upper half), $^{31}$P NMR (top, lower half), and $^{31}$P$\{^1$H$\}$ NMR (bottom, lower half) spectra of **1b** (c = 0.5 mol/L) after stirring at r.t. for 21 d in the presence of 5 mol% of [($\eta^5$ :$\eta^1$ -C$_5$H$_4$C$_{10}$H$_{14}$)$_2$Ti], Figure S20: $^1$H NMR spectrum of neat **1c** after stirring at r.t. for 4 d, Figure S21: $^{11}$B NMR (top, upper half), $^{11}$B$\{^1$H$\}$ NMR (bottom, upper half), $^{31}$P NMR (top, lower half), and $^{31}$P$\{^1$H$\}$ NMR (bottom, lower half) spectra of **1c** (neat) after stirring at r.t. for 4 d, Figure S22: $^1$H NMR spectrum of neat **1c** after stirring for 16h at 323 K, Figure S23: $^{31}$P NMR (top) and $^{31}$P$\{^1$H$\}$ NMR (bottom) spectra of **1c** (neat) after stirring for 16h at 323 K, Figure S24: $^{11}$B NMR (top) and $^{11}$B$\{$1H$\}$ NMR (bottom) spectra of **1c** (neat) after stirring for 16h at 323 K Figure S25: $^{31}$P$\{^1$H$\}$ NMR spectra of **1c** (c = 0.4 mol/L) after stirring for 21d (a), 7d (b), 16h (c), 90 min (d), 30 min (e) at r.t. in the presence of 4 mol% of [($\eta^5$ :$\eta^1$ -C$_5$H$_4$C$_{10}$H$_{14}$)$_2$Ti], Figure S26: $^{31}$P$\{^1$H$\}$ NMR spectra of **1c** in toluene at.r.t for 3h under different conditions: a) 5 mol% [Ti], c (**1c**) = 0.1 mol/L; b) 10 mol% [Ti], c (**1c**) = 0.1 mol/L; c) 10 mol% [Ti], c (**1c**) = 0.1 mol/L, in 1:1 mixture of THF and toluene; d) 10 mol% [Ti], c (**1c**) = 0.2 mol/L; e) 25 mol% [Ti], c (**1c**) = 0.1 mol/L; f) in absence of [Ti], c (**1c**) = 0.1 mol/L, Figure S27: $^1$H NMR spectrum of isolated poly-**1c** in C$_6$D$_6$, Figure S28: $^{31}$P NMR (top) and $^{31}$P$\{$1H$\}$ NMR (bottom) spectra of isolated poly-**1c** in C$_6$D$_6$, Figure S29: $^{11}$B NMR (top) and $^{11}$B$\{^1$H$\}$ NMR (bottom) spectra of isolated poly-**1c** in C$_6$D$_6$, Figure S30: $^{31}$P$\{^1$H$\}$ NMR spectra of 2 in toluene after 3h at r.t. in the absence of [Ti] (bottom), in the presence of 5mol% [Ti] (middle) and in the presence of 10 mol% [Ti] (top)

**Author Contributions:** Conceptualization, methodology; investigation, writing original draft, preparation, visualization, F.L.; resources, writing, review and editing, supervision, project administration, funding acquisition, M.S.; Delivering Ti catalyst, T.O. and R.B. All authors have read and agreed to the published version of the manuscript.

**Funding:** This research was funded by the Deutsche Forschungsgemeinschaft within the project Sche 384/41-1.

**Data Availability Statement:** Data are available upon reasonable request from the authors.

**Conflicts of Interest:** The authors declare no conflicts of interest.

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
