# Peer review of "Mono-Alkyl-Substituted Phosphinoboranes (HRP–BH2–NMe3) as Precursors for Poly(alkylphosphinoborane)s: Improved Synthesis and Comparative Study"

_inorganics, doi:10.3390/inorganics11100377_

Round 1

Reviewer 1 Report

The authors report in this manuscript the synthesis of a series of alkyl-substituted phosphinoboranes. Although the experimental work is well carried out, I wonder about the motivation of the work. I do not see a strong motivation for the synthesis of these compounds.

According to some previous papers on phosphine boranes, the merit of this type of compound is that they can provide H2 from mild themolysis (for use in hydrogen release similar to ammonia boranes). So a question of interest is about the H2 release capacity of these compounds. Does the amino group reinforce this ability?

In addition, the authors stated about a "comparative study", it is not clear about the real improvement of this synthetic approach with respect to the well known synthetic procedures for making phosphino. It is certainly different, but is it really improved. The authors need to make a convincing proof thatit is the case.

Author Response

According to some previous papers on phosphine boranes, the merit of this type of compound is that they can provide H2 from mild thermolysis (for use in hydrogen release similar to ammonia boranes). So a question of interest is about the H2 release capacity of these compounds. Does the amino group reinforce this ability?

Response: Unfortunately not, however we did not tried it by using a usual H2-elimination catalyst as for ammonia boranes developed. That would be a good idea for further studies, many thanks for this hint! H2-elimination without a catalyst only occurs, according to our experiences in a pentacoordinated transition state on the group 13 element, which was shown to work for corresponding LA/LB stabilized phosphanylalanes and -gallanes (J. Organomet. Chem. 2006, 691, 4556 – 4564). For such LB stabilizes pentelylboranes we never observed it. But with a catalyst, lets try!

In addition, the authors stated about a "comparative study", it is not clear about the real improvement of this synthetic approach with respect to the well known synthetic procedures for making phosphino. It is certainly different, but is it really improved. The authors need to make a convincing proof that it is the case.

Response: The reported pathway is special to receive primary phosphines. The steps afterwards (metalation and reaction with iodoboranes) we used already. In fact primary phosphines are obtained by metalation of PCl3 and subsequent reduction with LiAlH4, in a two step synthesis or sometimes by hydrophosphination of PH3. We found, using NaPH2 as source for primary phosphines by adding commercially available electrophiles is easier than preparing arbitrarily alkyl nucleophiles (hard as Li or alkalimetal salts, easier as Grignard reagents), since usually the stoichiometry of the reagents is not right and some dimetallation occurs which needed to be purified of the resulting primary and secondary chlorophosphines by distillation, resulting in low yields. The current method, by using a H-shape Schlenk-flask, is a one-pot reaction to give pure primary phosphines and just in a second step mono-alkyl substituted phosphanylboranes in gram scale and good yields. The reactions can be upscaled to a large scale.

Reviewer 2 Report

An excellent work. A workable on-pot way of making hitherto difficult to access mono-alkyl-substituted phosphinoboranes HRP–BH2–NMe3 has been developed. Polymerisation has been studied, both catalytic and thermal. This work is an important step towards new materials with B-P inorganic skeleton. 

Author Response

Many thanks for this very encouraging comment. We work hard to improve the knowledge of the community. 

Reviewer 3 Report

Manfred Scheer and co-authors developed a new approach to mono-alkyl-substituted phosphinoboranes for further utilization of the synthesized compounds in polymerization. The target compounds were obtained from alkyl halides and NaPH2 followed by metalation and salt metathesis. Indeed, previously described procedures for alkyl-substituted phosphinoboranes are complicated and inconvenient, so, the new pathway will be interesting for the readership. I found the paper useful and interesting, so it can be published in Inorganics journal as is. Before publishing it would be helpful to answer the question about any useful properties of the synthesized polymers. Also, Figure 4 may be placed in SI, since it is a ESIMS-spectrum.

Author Response

The properties of these polymers are currently under investigation. Thus the current results are a first step into this area. Concerning Figure 4: we would like to stay with it in the main text, since it shows the end-group variations of the formed polymer and therefore is important to be reported in the main text.